# Environmental and Motivational Determinants of Physical Activity among Canadian Inuit in the Arctic

**DOI:** 10.3390/ijerph16132437

**Published:** 2019-07-09

**Authors:** Victor O. Akande, Robert A.C. Ruiter, Stef P.J. Kremers

**Affiliations:** 1Department of Health Promotion, Maastricht University, NUTRIM School of Nutrition and Translational Research in Metabolism, Maastricht University Medical Center+, P.O. Box 616, 6200 MD Maastricht, The Netherlands; 2Department of Work & Social Psychology, Faculty of Psychology & Neuroscience, Maastricht University, Universiteitssingel 40, 6200 MD Maastricht, The Netherlands

**Keywords:** Arctic, Inuit, environment, active, steps, pedometer, determinants, regulation, and promotion

## Abstract

Background: Canadian Inuit have transited from a physically active hunter-gatherer subsistence lifestyle into sedentary ways of life. The purpose of the current study was to measure physical activity levels among Nunavut Inuit adults, and explore the socio-cognitive and environmental factors influencing the number of steps taken per day. Method: Inuit and non-Inuit adults (*N* = 272) in Nunavut participated in a seven-day pedometer study during summer and winter seasons. Participants were asked to complete the Neighbourhood Environmental Walkability Scale (NEWS) and Behavioral Regulation in Exercise Questionnaire (BREQ-3). Data analyses included descriptive statistics, hierarchical linear regression, and tests of mediation effects. Results: Participants had limited to low activity at a rate of 5027 ± 1799 and 4186 ± 1446 steps per day, during summer and winter, respectively. There were no seasonal and age effects on the number of steps. Gender effects and community differences were observed. Perceived infrastructure and safety as well as land use mix diversity were found to be positive environmental correlates of steps taken, which were partially mediated by identified motivational regulation. Conclusion: Physical activity levels among Nunavut adults are generally low, but can be promoted by improving the external physical environment and internal motivational regulation.

## 1. Introduction

The high prevalence rates of chronic diseases such as Type 2 diabetes, cardiovascular diseases and certain cancers among Nunavut Inuit have been linked to a shift from an extremely physically active nomadic lifestyle to technology-driven sedentary ways of life [1]. These trends have led to a growing call by public health experts to focus efforts on behavioural interventions to promote active living in the population. Nunavut Inuit have experienced rapid social, cultural and environmental changes following contacts with European immigrants over the past decades. The changes in many communities have resulted in significant disruption to traditional ways of life and erosion of cultural practices and values [2,3,4]. Nowadays, modernization and technology-driven approaches such as consumption of store-bought processed foods, motorized transportation and white-collar jobs have largely replaced hunting, fishing, and other traditional subsistence activities of the pre-contact era that involved significant daily energy expenditures [4]. Evidence suggests that these adopted Eurocentric ways of life have significantly eroded the moderate to high levels of physical activity and fitness among Nunavut Inuit resulting in weight gain and attendant health issues. Findings from a longitudinal study in Nunavut [5] indicated decreases in physical activity, aerobic power and muscle strength among Nunavut Inuit over a period of 20 years. The study also found increases in subcutaneous fats and higher body mass index for men and women 40 years and over within the period of the study. While obesity rates and associated chronic diseases have increased in the last 25 years, there has been a dearth of studies that examined physical activity levels in the population. To the best of our knowledge, the current study is the first to report objectively measured physical activity levels among Nunavut adults and the influence of environmental factors and internal motivational regulations. 

### 1.1. Environment and Walking Behaviour 

There is a rapidly growing body of empirical evidence in support of the influence of environmental factors on physical activity [6]. The built environment is underpinned by attributes that influence physical activity behavior [6,7,8]. This implies that modifications to the built environment may promote physical activity participation or foster sedentary behaviours and the attendant adverse health effects. In Nunavut, changes to the built environment are an integral part of the social, cultural and environmental shifts that the Canadian Arctic has experienced over the past five decades. The changes are correlated with decreased physical activity participation according to research evidence [8]. Particularly, subsistence activities that were associated with walking behaviour, such as hunting, fishing, traditional food processing, etc., have remarkably decreased [4]. There is empirical evidence in support of a positive relationship between community walkability and residents’ walking behaviour [9,10,11]. 

Globally, and particularly in the developed world, the number of steps taken per day is widely used as a reliable measure of physical activity levels in both adults and children [12,13,14,15]. Tudor-Locke and colleagues developed a graduated step index for analyzing the number of steps taken as an indicator of physical activity levels [16,17]. According to the index, healthy adults who take ≤ 2500 steps/day are considered as operating at basal activity level; 2500–4999 steps/day is described as having limited activity; 5000–7499 steps/day is low activity; adults who take 7500–9999 steps/day are classified as somewhat active; 10,000–12,499 steps/day are active; and ≥ 12,500/day is rated as highly active. This graduated scale allows experts to objectively determine what fraction of a population is actually physically active. Additionally, to derive optimal health benefits from the number of steps taken, researchers have associated the number of steps taken with positive health outcomes. A study of 93 menopausal women revealed that participants who took 5000–7500 steps/day had significantly lower body mass index compared to their counterparts who took less than 5000 steps/ day [18]. In the same linear relationship, participants who took 7500–9900 steps/day had remarkably lower BMI in contrast to those who took 5000–7500 steps/day. However, no significant differences were found between those who took 7500–9900 steps/day and those who took over 9900 steps/day. A similar positive relationship was found in women with depression. According to McKercher et al. [19], women who took ≥ 7500 steps/day have been associated with about 50% decreases in depression compared with those who took < 5000 steps/day. Men who achieved ≥ 12,500 steps/day also reported 50% reduction in prevalence rate for depression. 

Taken together, the findings suggest that a certain minimum number of steps among adults are linked to certain health benefits. Findings from a meta analysis conducted by Bohannon [20] on step-defined physical activity measured by accelerometers and pedometers provided an average number of steps taken by residents of some countries and suggested that physical inactivity is a global problem [14,15]. For example, adults in the United States take approximately 5100 steps/day [21]; in Japan, the number of steps taken by 15 years and older are 7200/day [22]; in Belgium: 9600 steps/day for 25–75 years old [23]; in Western Australia: 9600 steps/day [24]; in Switzerland: 25–74 years old took 10,400 and 8900 steps/day for men and women, respectively [25]. In Canada, men accumulate 9500 steps per day and women, 8400 steps/day [13]. However, there is a dearth of reliable data about the number of steps taken by the Nunavut Inuit adult population.

### 1.2. Motivational Regulation

Scholars have proposed to integrate social psychological factors in ecological approaches to determinant studies in order to increase our understanding of the underlying processes and mechanisms leading to behavior change [26,27,28]. An example of a social psychological theory that is widely used in predicting self-regulated behaviors such as physical activity is Self-Determination Theory (SDT) [26]. The theory is predicated on the notion that the quality of an individual’s motivation influences whether the individual will engage in, and sustain, the health behaviour [26]. SDT distinguishes *six motivational regulations* that reside along a continuum from *amotivation*, via controlled motivation (i.e., *external* and *introjected regulation*), to autonomous motivation (i.e., *identified regulation*, *integrated regulation* and-the optimal form-*intrinsic motivation*). Controlled motivation is characterized by feelings of pressure and lack of choice, either emanating from factors situated outside the person (e.g., punishment, deadlines: *external regulation*), or from inside factors (e.g., guilt, shame: *introjected regulation*). Autonomous motivation, as opposed to controlled motivation, is characterized by experiencing a sense of freedom in one’s choices and is driven by feelings of personal relevance (e.g., exercise is important to me because it is good for my health: *identified regulation*), personal identity (e.g., I am a sporty type: *integrated regulation*) or enjoyment (*intrinsic motivation*) [26]. Previous studies have demonstrated that autonomous motivation is associated with more favourable outcomes (e.g., greater well-being, greater participation in physical activity in a variety of contexts and greater perseverance [28,29]. Thus, a higher environmental walkability may increase an individual’s motivation for exercise, for example the perception of easier walking opportunities may lead to increased perceived behavioural control [27]. 

The present study aimed to objectively assess the levels of physical activity among Nunavut Inuit adults by determining the average number of steps taken per day. In addition, we explored the environmental walkability and motivational factors influencing the number of steps taken. Based on previous empirical studies [8,11] and theory [e.g., the Environmental Research framework weight Gain prevention (EnRG; [27,30]), it was expected that environmental walkability factors would be positively associated with the number of steps taken, and these associations would be partially mediated by motivational regulation (Figure 1). To the best of our knowledge, this was the first study that addressed environmental factors and motivational determinants of objectively assessed physical activity levels among adults in Nunavut.

## 2. Materials and Methods

### 2.1. The Canadian Inuit in the Arctic

The Inuit people are one of the three Indigenous/Aboriginal peoples in Canada. The Inuit inhabit *Nunaat*, *Inuktitut* language expression for traditional Inuit homeland, which spans across the Canadian Arctic, occupying more than one-third of Canada’s land mass. *Nunaat* spreads across four discrete geographic regions—Inuvialuit, Nunavik, Nunatsiavut and Nunavut [31]. Nunavut is the largest of the four regions, becoming a territory in 1999. The area has a population of 37,082 (84.2% Inuit and 15.8% non-Inuit) and is divided into three regions: Qikiqtaaluk (North and South sub-regions), Kivalliq, and Kitikmeot. The Inuit have the highest population growth rate and are the youngest demographic group in Canada with a median age of 22, compared to 40 nationally [32]. Four communities in Nunavut were selected from the three regions: Resolute Bay (North Qikiqtaaluk), Iqaluit (South Qikiqtaaluk), Cambridge Bay (Kitikmeot), Baker Lake (Kivalliq) (see Table 1). 

### 2.2. Community Profile

Seven communities were initially contacted and provided with information on the purpose of the study, including potential benefits to the community and Nunavut in general. Four communities were selected based on the letters of support that were received from community leaders/administrators, the population size, and geographic spread across the three regions. The recruitment procedure strived for participation from the most populated (Iqaluit) to one of the least populated communities (Resolute Bay) in the territory. Demographic descriptions of the participants in the four participating communities are presented in Table 2. Approval for the study was obtained from the Ethics Review Committee for psychology and neuroscience at Maastricht University, Netherlands (reference number ECP-148 05_03_2015), and from the Nunavut Research Institute (License Number 050 1315-Amended). 

### 2.3. Psychometric Measures

#### Neighborhood Environmental Walkability Scale

In this study, residents’ perceptions of community walkability were obtained by administering the Neighbourhood Environmental Walkability Scale (NEWS), a self-reported perceived environment survey that was originally developed by Saelens and colleagues [9]. Our study utilized the Confirmatory Factor Analysis (CFA)-based NEWS, which comprised of eight multi-item and five single-item subscales. The CFA-based data analysis is predicated on the scoring algorithms and procedures fully described in Cerin et al [33]. We utilized the NEWS instrument to assess residents’ perception of neighbourhood attributes in relation to the number of steps taken per day. Items were grouped into subscales as previously described [10] to evaluate the underlying constructs of land-use mix diversity and land-use mix access, perceived safety from traffic and crime, proximity to stores, schools, and other facilities, residents’ perceived access to these destinations, perceived levels of infrastructure and facilities for walking, cycling, and neighbourhood aesthetics. 

### 2.4. Motivational Regulation of Exercise

To assess controlled and autonomous motivational regulation for participation in physical activity and determine whether the effect of the environmental factors on the number of steps taken was mediated by motivational regulation, we administered the Behavioural Regulation in Exercise Questionnaire (BREQ-3), originally developed by Mullan and colleagues [34]. BREQ is a multi-dimensional 24-item instrument that measures stages of self-determination continuum regarding respondents’ motivation to exercise based on a five-point Likert scale (ranging from 1: not true for me, to 5: very true for me). Six constructs/factors of motivational regulation were measured (intrinsic regulation, integrated regulation, identified regulation, introjected regulation, external regulation, and amotivation), and each as a subscale consisting of four items. The score for each factor was determined as the average value of four items that constituted each subscale/factor. 

### 2.5. Behavior

Participants were asked to wear the Kaden G-Sport Pocket Pedometer 793 Multi-function step/distance/calories/ counter. The participants were asked to wear the pedometer for step counting purposes by attaching the instrument to their waists after dressing up in the morning, ready to proceed with the day’s activities. The pedometer was to remain attached to the waist until the participants were ready to head to bed in the evening. This provided an average of approximately 13 hours of wear time per participant. Each participant was required to record the reading just before heading to bed, without altering or resetting the pedometer reading. This was done daily for seven consecutive days. This way, the readings for the seven days were added in a cumulative manner. The readings were cross-referenced with the daily seven daily readings that were recorded by the participants. There were no discrepancies between the two readings. The average of the seven-day readings was determined by adding seven readings together and dividing the value by seven, to provide the average number of steps taken per day. The seven-day readings on the pedometers were then used to confirm the reports provided by the participants. 

### 2.6. Pilot Study

Both questionnaires (NEWS and BREQ) were pilot-tested in Iqaluit among the target population that varied across age, gender, ethnicity, educational, and socioeconomic class. Results from the pilot test necessitated a few changes to the wordings of some items to enable the research participants to better understand the study and increase the reliability of data by ensuring validity of the assessment of the various constructs that we explored. Some NEWS items were rephrased as suggested by participants who participated in the pilot study. Given the geographical location of Nunavut and some unique environmental factors that are not captured by the NEWS instrument, we developed a three-item “weather conditions” scale to measure the respondents’ perceptions of the weather conditions. The three items were: (1) the weather conditions (wind, storm) make it difficult for me to engage in walking or cycling in my neighborhood; (2) the weather conditions (blizzard, snow, poor visibility) make it difficult to for me to engage in walking or cycling; and (3) the seasons (winter, long cold season) make it difficult for me to engage in walking/cycling/other outdoor activities).

Given the lack of documented evidence on prior use of the Kaden G-Sport Pedometer (Kaden Group, Zhuhai, China) in physical activity studies, we recruited 12 regular users of pedometers to use Kaden G-Sport brand simultaneously with their conventional pedometers (Yamax brands) over a seven-day period. The purpose was to determine the reliability of the Kaden G-Sport instrument. In terms of the number of steps, the overall difference between the two instruments was between 2-4%, indicating that the reliability of the Kaden G-brand is similar to validated brands such as the Yamax’s. 

### 2.7. Data Collection—Main Study

To determine the number of steps taken per day in the summer (July/August 2016), 153 males and 119 females (*n* = 272), both Inuit and non-Inuit healthy adults 18 years old and over (mean age: 34.92, SD: 12.62), were selected through a random sampling approach from cross sections of the four communities. Inuit and non-Inuit who were less than 18 years of age, not in good state of health, or adults who were above 65 years of age, were excluded from the study. The representations of Inuit versus non-Inuit were made to be as close as possible to the population distribution of the two groups in each of the four communities. Each respondent was provided with a pedometer and asked to complete the NEWS and BREQ surveys. Research assistants provided information on the study to research participants and obtained written consents prior to the start of the study. Research assistants also provided necessary supports and clarifications as needed, including guidance on the use of the pedometers, and ensured that the forms were properly completed. The pedometer study was repeated with the same participants in the winter (January 2017), to assess the number of steps taken, and determine if there was a seasonal effect that resulted in change in the number of steps taken from summer to winter (delta steps). Pedometers were retrieved from the participants after seven days, and the steps data retrieved. Step data that were less than seven days were excluded from the analysis. Of the 272 participants who completed the survey in the summer, only 169 (62%) participated in the pedometer study. The number dropped to 148 participants (12.5% attrition) in the winter when the pedometer study was repeated. No age, ethnicity and community level differences were observed in attrition rates. 

### 2.8. Data Analysis

The collected data were entered in IBM SPSS Statistics Version 24 (SPSS Inc., Chicago, IL, USA) for cleaning and subsequent analyses. Data were analyzed using descriptive statistics, Pearson correlations, hierarchical linear regressions, and test of mediation effects. Five missing values were input by item means. Descriptive analyses included the mean and standard deviation of score values. Subscale scores were computed by summing the scores on the respective subscales: the higher the score, the more agreeable the respondent, for instance, the motivation for engaging in exercise. 

Internal consistency analyses of the six BREQ subscales revealed good psychometric properties according to their Cronbach alpha (α) values: Amotivation = 0.65; external regulation = 0.75; introjected regulation = 0.76; identified regulation = 0.63; intrinsic regulation = 0.82; and integrated regulation = 0.82. To determine whether the NEWS subscales were reliable, their Cronbach α values were assessed. For Infrastructure subscale, α = 0.79. Land-use mix access subscale was α = 0.51, indicating poor internal consistency. However, elimination of one item “it is easier to walk to a transit stop (bus, train) from here” increased the reliability score to 0.68. This concerted the factor to a three-item subscale. The subscales of street connectivity (0.55) and aesthetics (0.59) had weak internal consistency. Elimination of items with the least contribution to the internal consistency of these subscales did not significantly improve the reliability score. Thus, the original subscales were retained. An assessment of the internal consistency reliability of the newly developed weather scale indicated good psychometric properties at a Cronbach α value of 0.83. The Cronbach α values of traffic hazard (0.10) and crime (0.18) were too low, and consequently removed from further analyses. 

Scores for the study variables were checked for normal distributions using tests for skewness and kurtosis [35]. Pearson correlations were computed to explore relationships among study variables. Further, tests for regression diagnostics for outliers were conducted as recommended by Fox [36]. To determine the validity of the hypothesis that motivational regulation would mediate the effect of environmental factors on the number of steps taken (summer, winter and delta steps) as a measure of physical activity, we utilized a hierarchical linear regression model. In Stage 1, all the environmental factors (NEWS subscales) were entered as predictor variables and linearly regressed against the number of steps taken in the summer by eliminating the least contributing variables to the model. We then determined the fraction of the variance in the number of steps taken that was explained by the environmental factors in the final model of Stage 1, while controlling for demographic factors (age, gender, ethnicity). In Stage 2, the six BREQ subscales were added to the model. The final predictor variables (environmental and motivational factors) were identified in the final model as well as the fraction of the variance in the number of steps explained by each. Stages 1 and 2 were repeated for each of winter and delta steps (winter–summer). To determine whether there was a mediation effect on the number of steps taken, we conducted mediation analysis according to MacKinnon et al 2007 [37], using the Bootstrap Confidence Intervals as described by Preacher & Kelly, 2011 [38].

## 3. Results

The Mean (SD) of the summer, winter and delta steps taken per day by respondents were 5027 (1798), 4186 (1445), and 835 (572), respectively. Although there were no significant differences between the overall means of summer and winter steps, further analysis using Tudor Lock’s graduated step index revealed some moderate to significant differences between summer and winter steps at some levels of graduation (Table 3). A higher proportion of participants engage in walking activities in the summer at steps index ≥ 5000 steps/day, when compared to winter. The data also suggest that, independent of seasons, significantly fewer people participate in walking activities at levels classified as “somewhat active” to “highly active”, that is ≥7500 steps/day.

The gender effect was consistent over summer and winter seasons. Men were more active than women. Additionally, people living in Iqaluit were more active than those living in Baker Lake, Cambridge Bay, and Resolute Bay. 

In Stage 1 of the hierarchical regression model (Table 4), two environmental factors were significantly and positively associated with physical activity in summer and winter: infrastructure/safety and land use (land-use mix diversity). In Stage 2, after adding the six motivational factors to the model, our findings indicated that identified motivational regulation added to the explained variance of the model in both summer and winter. No other motivational factors had any significant observed regulatory effect. 

The impact of identified regulation appears to be consistent over the summer and winter. When predicting the change in steps from summer to winter, only infrastructure was a significant predictor, indicating that good infrastructure leads to increases in the number of steps taken by respondents. No significant associations for motivational factors were observed for the change in steps (Table 4). The explained variance of this final model was substantially lower (12%) than in summer (44%) and winter (40%).

### Mediation Analysis

Results from the correlation studies and regression analyses indicated that only two independent variables were of significance in the study: Infrastructure/safety and land use mix diversity, and only identified motivational regulation was significantly associated with the number of steps taken in both summer and winter seasons (Table 4). 

To explore the mediation effect of identified regulation, we hypothesized that, (1) there was a direct effect of two independent variables, infrastructure/safety and land use mix diversity, on the number of steps taken in both summer and winter seasons. That is, there would be a correlation between each of the two independent variables and the number of steps taken; (2) there would be a correlation between each of the two independent variables and identified regulation; (3) there would be a correlation between identified regulation and the number of steps taken while controlling for each of the two independent variables, and (4) the effect of the independent variables on the number of steps taken while controlling for identified regulation would be zero, indicating a full mediation. The results of mediation analysis indicated a strong relationship between each of infrastructure/safety and land use mix diversity, and identified regulation in the winter season. There was a direct effect of each of the two variables on the number of steps taken, and there was an indirect (mediated) effect.

X is the independent variable (infrastructure/safety or land use mix diversity); Y is the number of steps taken in either winter or summer; and M is the mediator variable, identified regulation (Table 5).

The analyses showed partial mediation by identified regulation given that the direct effect, that is, the effect of the infrastructure/safety while controlling for identified regulation does not equal zero (Table 5). Similar effects were observed in the summer for the infrastructure/safety variable. In contrast, no meaningful relationship was observed between land use mix diversity and identified regulation, and no significant indirect effects were observed between land use mix diversity and the number of steps taken in the summer (the Bootstrapped CIs for the indirect effect and the association between X and M include zero). Note that the associations between infrastructure/safety and identified motivation were negative.

## 4. Discussion

The study determined the number of steps taken per day by Nunavut residents as a measure of physical activity in the Canadian Arctic and explored the perceived environmental walkability and motivational regulation as correlates of physical activity in four communities. Our findings showed that only 2.7% of Nunavut residents were physically active at ≥10000 steps/day in the summer (and none in the winter) compared to about 35% of adult Canadians who live elsewhere in the country [13]. The sharp disparity in the physical activity levels between residents of Nunavut and the rest of Canada may be associated with unique environmental attributes in the Canadian Arctic in contrast to the southern part of Canada. These attributes are probably related to the weather. Since this is a non-changeable environmental determinant of physical inactivity, insights into the role of the built environment in Nunavut in terms of its density, diversity and design are even more important [39]. We found that perceived infrastructure and safety, as well as land-use mix were found to be positive environmental associates of steps taken. In addition, our findings suggest that Iqaluit residents are more physically active than residents of Baker Lake, Cambridge Bay and Resolute Bay. The observed differences may be explained in part by dissimilarities in the built environment. Evidence suggests a link between increased density and land-use mix, and walking for both pleasure and travel [40]. 

Iqaluit is more densely populated, has better street connectivity, land-use mix, and land use mix access and aesthetics, as well as improved community infrastructure, including good roads, street lighting, and fitness centers, compared to the other three communities. When predicting the change in steps, only infrastructure was a significant predictor, indicating that good infrastructure leads to a smaller decrease in the number of steps taken by residents when transitioning from summer to winter conditions. In general, as community land-use mix increases, there is an increased probability that there will be desirable or useful destinations that are in close proximity of one another, that may potentially motivate individuals to walk, rather than drive, to access those destinations. This appears to be the case for Iqaluit in contrast to the other communities. Environmental temperature may also have played a factor as Iqaluit appears to be relatively warmer by five to six degrees in the winter.

In general, our results were consistent with previous findings: improved infrastructure and better land use (land-use mix diversity) were significantly associated with physical activity in both summer and winter seasons. In a sample of 351 Canadian adults, Rhodes and workers [41] found that residing in a walkable neighbourhood tended to increase accelerometer-based minutes of cycling and recreational walking and less of motorized transport. In another study involving Australian adults, neighbourhood walkability was positively correlated with frequency of walking as a means of transportation, although the same effect was not found for recreation walking [42]. Additionally, people who live in neighbourhoods with higher residential density, street connectivity, better land-use mix in the United States were found to be more likely to walk and cycle than in neighbourhoods with less of these environmental characteristics [9].

We also explored the contributions of motivational regulation in the observed effects of the environmental factors on the number of steps taken. Identified regulation made a significant contribution to the model. Our findings indicated that environmental effects were partially mediated by identified motivational regulation, especially in the winter season. Identified regulation is an autonomously driven form of extrinsic motivation; it reflects the conscious evaluation of physical activity and the perceived benefits. More autonomous forms of motivational regulation (i.e., intrinsic and integrated regulation) were not significant correlates of physical activity. This may be viewed as a surprise, also in the light of previous empirical investigations [43,44,45,46] that have shown consistent and positive impact of these types of motivation in predicting physical activity. It seems that the Arctic context may hamper the translation of intrinsic and integrated motives (for example, engaging in leisure time sports for enjoyment) into action. The impact of identified regulation in the Arctic context underlines the importance of communication about the (health) benefits of physical activity. These findings provide some foundational work for future studies that aim to identify evidence-informed psychological constructs that are viable and appropriate targets for behaviour change interventions [47,48,49,50] to promote increased physical activity in the Canadian Arctic. 

It is also important to note that the proportion of the direct effect of environmental walkability on the number of steps was negative. In our data, bivariate correlations between infrastructure/safety and identified motivation, as well as between land use mix diversity and identified motivation were positive (r’s 0.11 and 0.30, respectively). Correlations between infrastructure/safety and number of steps, as well as between land use mix diversity and number of steps were positive as well (r’s between 0.26 and 0.42 in summer and winter). Bivariate correlations between identified motivation and number of steps were also positive (0.22 in the summer and 0.26 in the winter). It is remarkable that when two predictor variables are positively correlated, and both of them correlate positively with the outcome variable, that the relationship between a predictor and an outcome in a multivariate analysis is negative. When this occurs, the hypothesised mediator acts as a suppressor variable and the mediation process is often referred to as inconsistent mediation [49]. This model can be differentiated from consistent mediation models, a situation in which the direct and mediated effects have the same sign. The multivariate results might indicate that the worse the environmental walkability, the more the individual needs identified motivational regulation to walk in that environment. And vice versa: the better the walkability, the less identified motivation is needed to walk. However, given the problematic nature of estimates from multivariate analysis, in our discussion of the results, we tried to comply with the recommendation that decisions on determinant associations and reflections on mediation be based on bivariate correlations [50]. Nevertheless, the results give amply room for new hypotheses and we recommend further studies on the intriguing relationships between relatively extreme environmental circumstances and strong culturally embedded beliefs such as those in the Arctic context.

### Study Limitations

The transferability of the study is limited due to very complex environment where the study was conducted. Although the study may be repeated with similar results/outcomes in other comparable environments, the complex Canadian Inuit environment may be an ingredient that makes this particular study interesting and require further studies to deepen our understanding of the impacts of the environmental conditions on lifestyle choices, particularly physical activity. Another potential limitation is the Kaden G-Sport instrument to assess physical activity. The instrument was not thoroughly validated given the small sample size of the participants used in the pilot. However, based on the very negligible differences in the number of steps taken between the two pedometers during the pilot, we would not expect a significant deviation from these results, if the sample size were larger and the pilot repeated in other environments.

## 5. Conclusions

According to a description of an effective behavioural change intervention mapping approach described by Kok et al [51], it is expedient to identify a determinant that predicts the behaviour and an effective method that can change the determinant. Such method should have practical applications and deemed culturally appropriate for the target population and context. The results of the current study indicate the importance of integrated approaches to the promotion of physical activity, focusing on improvement of infrastructure and land-use mix, in combination with educational interventions addressing the health benefits of being physically active in Nunavut.

## Figures and Tables

**Figure 1 ijerph-16-02437-f001:**
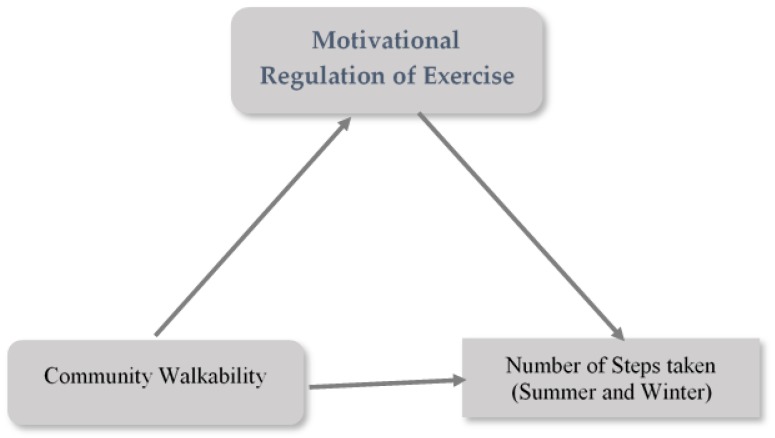
Research model.

**Table 1 ijerph-16-02437-t001:** Demographic and environmental characteristics of the four participating communities.

Community	Population	% Male	% Female	% Non-Inuit	% Inuit	Average Temperatures		Average Snow
						Jan. (Winter)	Aug. (Summer)	Jan./Aug.
Baker Lake	1997	54	46	9.1	90.9	−33.3	+9.8	32 cm/0 cm
Cambridge Bay	1746	50.5	49.5	20.7	79.3	−32.0	+6.8	24 cm/0 cm
Iqaluit	7590	51	49	44.6	55.4	−26.9	+7.1	22 cm/0 cm
Resolute Bay	210	58	42	14.3	85.7	−32.0	+2.0	20 cm/0 cm

**Table 2 ijerph-16-02437-t002:** Demographic Profiles of research participants (sample size *n* = 272).

Variable	Sample Size (*n*)	%
Age		
18–29	117	43.0
30–39	57	21.0
40–49	59	21.7
50–64	39	14.3
Gender		
Male	153	56.2
Female	119	43.8
Ethnicity		
Inuit	203	74.6
Non-Inuit	69	25.4
Community		
Baker Laker	62	22.8
Cambridge Bay	48	17.6
Iqaluit	139	51.1
Resolute Bay	23	8.5

**Table 3 ijerph-16-02437-t003:** Application of Tudor Locke’s graduated index to the number of steps taken by research participants in summer and winter.

Graduated Steps Index	Summer Steps (%)	Winter Steps (%)
<2500 steps/day	3.4	13.4
2500–4999 steps/day	50.7	64.5
5000–7499 steps/day	35.1	18.0
7500–9999 steps/day	8.1	4.1
10,000–12,499 steps/day	2.7	0

**Table 4 ijerph-16-02437-t004:** Regression coefficients, *p*-values and explained variance from regression analysis for the number of steps taken during the summer and winter seasons, and the change (delta) in the number of steps taken from summer to winter (*n* = 272).

Variables	Summer Steps						Winter Steps						Delta Steps					
	β	*p*-Value	R^2^	β	*p*-Value	R^2^ Final Model	β	*p*-Value	R^2^	β	*p*-Value	R^2^ Final Model	β	*p*-Value	R^2^	β	*p*-Value	R^2^ Final Model
Age	−0.07	0.23	0.38	−0.07	0.35	0.44	−0.10	0.20	0.34	−0.07	0.36	0.40	−0.02	0.83	0.12	−0.02	0.84	0.12
Gender	−0.15	0.04		−0.14	0.04		-0.15	0.03		−0.15	0.02		0.00	0.99		0.00	0.99	
Ethnicity	0.00	0.98		−0.02	0.74		−0.02	0.82		−0.02	0.77		−0.04	0.65		−0.04	0.63	
Baker Lake	−0.01	0.87		0.07	0.36		−0.04	−0.06		0.05	0.53		0.08	0.36		0.09	0.31	
Cambridge Bay	−0.04	0.57		0.02	0.81		−0.05	0.44		0.01	0.89		0.00	0.99		0.01	0.94	
Resolute Bay	−0.40	<0.001		−0.41	<0.001		−0.41	<0.001		−0.16	0.06		−0.16	0.06		−0.16	0.06	
Land−Use Mix Diversity	0.20	0.01		--	--		0.19	<0.001		--	--		--	--		--	--	
Infrastructure & Safety	0.25	<0.001		0.27	<0.001		0.23	<0.001		0.23	<0.001		0.25	<0.001		0.25	<0.001	
Identified Regulation	--	--		0.25	<0.001		--	--		0.27	<0.001		--	--		--	--	

**Table 5 ijerph-16-02437-t005:** Partial mediation effects of identified motivational regulation on the number of steps taken in the summer and winter seasons.

Indictors of Mediation	Unstandardized Effect Size and [Bootstrap Confidence Intervals]			
	Summer Season		Winter Season	
	Infrastructure/Safety	Land Use Mix Diversity	Infrastructure/Safety	Land Use Mix Diversity
Total effect	1342 [(888.17)–(1817.22)]	796 [(383.05)–(1209.12)]	960 [(596.13)–(1324.19)]	542 [(235.92)–(849.26)]
Direct effect of X on Y	1541 [(1065.11)–2017.21)]	855 [(443.03)–(1267.05)]	1129 [(767.61)–(1489.85)]	617 [(312.05)–(921.42)]
Indirect effect of X on Y	−199.5 [(−360.15)–(−67.05)]	−59 [(−0.170.25)–(6.61)]	−169 [(−293.12)–(−69.71)]	−74 [(−159.91)–(−12.06)]
Association between X and M	−0.463 [(−0.75260)–(−0.1733)]	−0.2023 [(−0.4464)–(0.0419)]	−0.463 [(-0.7024)–(-0.1699)]	−0.2441 [(−0.4624)–(−0.0258)]
Association between M and Y	430 [(169.94)–(691.84)]	291 [(17.96)–(565.00)]	386.49 [(184.92)–(588.86)]	303 [(93.54)–(513.88)]

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
