# Peer review of "Environmental and Motivational Determinants of Physical Activity among Canadian Inuit in the Arctic"

_ijerph, 2019, doi:10.3390/ijerph16132437_

Round 1

Reviewer 1 Report

This is a well constructed and performed research integrating different approaches to the issue of physical (in)activity. Also taking the group of Canadian Inuits seems to be interesting, not only because of the cultural/territorial difference from other Canadians but also as they can be treated as a native culture that has recently (last few decades) changed their behavior in terms of food and physical activity.

There are still some limitations. The transferability of the study is limited due to a very complex environment where the study was performed.

I also have some question to the authors:

1.     Was there any difference between age groups the study was done in a group 18-65 with the highest proportion of the young group 18-29. As I see it in Table 4 – there was no diff.

2.     Where there any data gathered about education level and BMI/obesity status in this group? Where there any diff?

Generally the study is very well designed and performed. The statistical methods are complex and in-depth with the data.

Author Response

REVIEWER 1

Comments and Suggestions for Authors

Point 1: This is a well constructed and performed research integrating different approaches to the issue of physical (in)activity. Also taking the group of Canadian Inuits seems to be interesting, not only because of the cultural/territorial difference from other Canadians but also as they can be treated as a native culture that has recently (last few decades) changed their behavior in terms of food and physical activity.

Response 1: Thank you for your compliments.

Point 2: There are still some limitations. The transferability of the study is limited due to a very complex environment where the study was performed.

Response 2: Thank you for your observation. We would also like to acknowledge the same limitations given the complex environmental conditions that Canadian Inuit are subjected to. However, although the study may be repeated with similar results/outcomes in other comparable environments, the complex Canadian Inuit environment may also be an ingredient that makes this particular study interesting.

I also have some question to the authors:

Point 3:   Was there any difference between age groups the study was done in a group 18-65 with the highest proportion of the young group 18-29. As I see it in Table 4 – there was no diff.

Response 3: Thank you. No significant effects of age were found on the number of steps taken during the study.

Point 4: Where there any data gathered about education level and BMI/obesity status in this group? Where there any diff?

Response 4: This is a good question. Information on education and the BMI status of participants were not included in the questionnaires in order to reduce participant burden to a minimum. In general, we can say that the population is minimally educated and overweight.

Point 5: Generally the study is very well designed and performed. The statistical methods are complex and in-depth with the data.

Response 5: Thank you for the observation.

Reviewer 2 Report

Below are my comments:

1.      Background and introduction are very clear to present the research objectives.

2.      Data collection method and measures of the study are clearly introduced.

3.      An imbalance sample size existed between Inuit and non-Inuit participants.

4.      Weekday and weekend steps should be analysed and contrasted besides the average steps calculated from the 7 days data.

5.      Kaden G-Sport pedometer could be a problem although it is validated to the Yamax’s pedometer. But the small sample size validation can be questionable.

6.      Please explain how the researchers deal with the step data which has less than 7 days?  

7.      Results and discussions are well organized and demonstrated the logical analysis of the research questions. 

Author Response

REVIEWER 2

Comments and Suggestions for Authors

Point 1:   Background and introduction are very clear to present the research objectives.

Response 1: Thank you.

Point 2:  Data collection method and measures of the study are clearly introduced.

Response 2: Thank you.

Point 3: An imbalance sample size existed between Inuit and non-Inuit participants.

Response 3: Thank you for your observation. The representations of Inuit versus non-Inuit in the study were made to be as close as possible to the population distribution of the two groups in each of the four communities.

Point 4: Weekday and weekend steps should be analysed and contrasted besides the average steps calculated from the 7 days data.

Response 4: Thank you for the suggestion. Unfortunately, we are unable to disaggregate the weekend data from the rest because participants were asked to choose any 7 consecutive days out of a 10-day period to complete the study, numbered as days 1-7. The data were therefore not recorded based on specific day of the week in all cases as Monday through Sunday.

Point 5: Kaden G-Sport pedometer could be a problem although it is validated to the Yamax’s pedometer. But the small sample size validation can be questionable.

Response 5: Thank you for your observation. We agree, the Kaden G-Sport has not been thoroughly validated given the small sample size of the participants used in the pilot. However, based on the very negligible differences in the number of steps taken between the two pedometers during the pilot, we would not expect a significant deviation from these results if the sample size were larger and the pilot repeated in other environments. 

Point 6: Please explain how the researchers deal with the step data which has less than 7 days?  

Response 6: Thank you. Step data that were less than 7 days were excluded from the analysis.

Point 7: Results and discussions are well organized and demonstrated the logical analysis of the research questions. 

Response 7: Thank you for your observation.

Reviewer 3 Report

Thank you for the invitation to review this manuscript. This study investigated the mediating effect of motivational regulation on the relationship between environmental walkability and the daily number of steps among Canadian Inuit in the Arctic. This topic is of high interest as it could bring perspectives in the preservation of the global health of these populations by reducing physical inactivity.

The introduction section is well written and interesting. However, a little more background on the relationship between the independent variable and the mediators (X to M) could be of interest in order to present the theoretical hypothesis behind the research model.

The methods section could also be slightly improved. In particular the mediation analysis part.  Could self-perceived motivational regulation impact self-perceived walkability of the environment? (reverse hypthesis?) Moreover, were there exclusion criteria and why no participants over 65 were included?

One of the main concerns is the interpretation of the mediation analysis section. The signs of the different effects in the mediation analysis were not interpreted by the authors. When interpreting the effects’ signs, the interpretation of the results seems very different from what the authors interpreted. Indeed, I would like to bring the authors’ attention to the signs of the indirect effect (always negative), and the sign of the direct effect (always positive).  Opposite signs of these effects usually mean “inconsistent mediation”. This seem to come from the relationship between walkability and motivation where high perceived walkability predicted low motivation but high motivation predicted high number of steps.  This lead to a discussion and conclusion section that are not supported by the results in my opinion. 

Author Response

REVIEWER 3:

Comments and Suggestions for Authors

Point 1: Thank you for the invitation to review this manuscript. This study investigated the mediating effect of motivational regulation on the relationship between environmental walkability and the daily number of steps among Canadian Inuit in the Arctic. This topic is of high interest as it could bring perspectives in the preservation of the global health of these populations by reducing physical inactivity.

Response 1: Thank you for your comments. 

Point 2: The introduction section is well written and interesting. However, a little more background on the relationship between the independent variable and the mediators (X to M) could be of interest in order to present the theoretical hypothesis behind the research model.

Response 2:  Thank you. We have included a statement describing the relationship between independent variable and the mediator in the introduction section. Please see lines 113-115 in the manuscript. Additionally, we now explicitly referred to the Environmental Research framework weight Gain prevention in order to assist the reader in recognising the theory behind our research framework.

Point 3: The methods section could also be slightly improved. In particular the mediation analysis part.  Could self-perceived motivational regulation impact self-perceived walkability of the environment? (reverse hypothesis?) Moreover, were there exclusion criteria and why no participants over 65 were included?

Response 3: Thanks for your comments. Yes, self-perceived motivational regulation could impact self-perceived environmental walkability. An individual who is self-motivated is less likely to have an unfavourable perception or feelings towards a harsher environment (with low walkability scores), thus making it easier to overcome such unfavourable environmental attributes and proceed to achieve his physical activity goals. We have now incorporated the reversed causation reflection in the Discussion section [lines 387-399].

Yes, there were exclusion criteria. Excluded criteria were Inuit and non-Inuit who were less than 18 years of age, not in good state of health, or adults who were above 65 years of age. Please see lines 221-222 in the manuscript.

Point 4: One of the main concerns is the interpretation of the mediation analysis section. The signs of the different effects in the mediation analysis were not interpreted by the authors. When interpreting the effects’ signs, the interpretation of the results seems very different from what the authors interpreted. Indeed, I would like to bring the authors’ attention to the signs of the indirect effect (always negative), and the sign of the direct effect (always positive).  Opposite signs of these effects usually mean “inconsistent mediation”. This seem to come from the relationship between walkability and motivation where high perceived walkability predicted low motivation but high motivation predicted high number of steps.  This lead to a discussion and conclusion section that are not supported by the results in my opinion. 

 Response 4: Thank you for the observation. Yes, there is inconsistent mediation. The proportion of the effect that is mediated can be negative in cases of inconsistent mediation as observed in our analysis. It is indeed possible that when two variables (environmental walkability and identified motivation) are positively correlated, and both of them correlate positively with the outcome variable (number of steps taken), that their relationship in a multivariate analysis is negative (Table 5). When this occurs, the mediator acts as a suppressor variable.

In our data, bivariate correlations between Infrastructure/safety and Identified motivation, as well as between Land use mix diversity and identified motivation were positive (r’s .11 and .30, respectively). Correlations between infrastructure/safety and number of steps, as well as between land use mix diversity and number of steps were positive as well (r’s between .26 and .42 in summer and winter). Bivariate correlations between identified motivation and number of steps were also positive (.22 in the summer and .26 in the winter). Thus, a situation in which the direction of the relationship between environmental walkability and the mediator variable, identified regulation, is reversed would indicate suppression. We have now added a sentence to the results section where we help the reader in identifying the negative association between X and M in the mediation analysis [lines 331-333].

Given the problematic nature of estimates from multivariate analysis, it is recommended that decisions on determinants association and mediation be based on bivariate correlations (Peters & Crutzen, 2017, p487-491). We are grateful for the observation of inconsistent mediation by the reviewer, leading us to a more in-depth discussion of this finding. Accordingly, we have discussed the contents of the relationships along the lines of the bivariate analyses, and we have incorporated a reflection on the inconsistent mediation that we found in the Discussion. Please see lines 387-399 in the manuscript.